# Inhibition of FGF and TGF-β Pathways in hESCs Identify STOX2 as a Novel SMAD2/4 Cofactor

**DOI:** 10.3390/biology9120470

**Published:** 2020-12-16

**Authors:** Peter F. Renz, Daniel Spies, Panagiota Tsikrika, Anton Wutz, Tobias A. Beyer, Constance Ciaudo

**Affiliations:** 1Department of Biology, Swiss Federal Institute of Technology Zurich, Institute of Molecular Health Sciences, Otto-Stern Weg 7, CH-8093 Zurich, Switzerland; peter.renz@uzh.ch (P.F.R.); spiesd@gmail.com (D.S.); g.tsikrika@gmail.com (P.T.); awutz@ethz.ch (A.W.); 2Molecular Life Science Program, Life Science Zurich Graduate School, Institute of Molecular Life Sciences, University of Zurich, Winterthurerstrasse 190, CH-8057 Zurich, Switzerland

**Keywords:** hESCs, pluripotency, TGF-β pathway, FGF pathway, time course differential expression analysis, STOX2

## Abstract

**Simple Summary:**

Signaling pathways are the means by which cells and tissue communicate, orchestrating key events during mammalian development, homeostasis, and disease. During development, signaling determines the identity of cells, and thereby controls morphogenesis and organ specification. Depending on the cellular context, these pathways can exert a broad range of even opposing functions. This is achieved, among other mechanisms, by crosstalk between pathways. Here, we examined how two pathways (the transforming growth factor-β (TGF-β) and the fibroblast growth factor (FGF)) cooperate in the maintenance and cell fate specification of human embryonic stem cells. We used inhibitory molecules for individual pathways on a short time series and analyzed the resulting variation in gene expression. In contrast to our expectations, we did not observe an extended crosstalk between the pathway at the gene regulatory level. However, we discovered STOX2 as a new primary target of the TGF-β signaling pathway. Our results show that STOX2 might act as a novel TGF-β signaling co-factor. Our work will contribute to understand how signaling by the TGF-β is mediated. In the future, these results might help to deepen our understanding of how signaling is propagated.

**Abstract:**

The fibroblast growth factor (FGF) and the transforming growth factor-β (TGF-β) pathways are both involved in the maintenance of human embryonic stem cells (hESCs) and regulate the onset of their differentiation. Their converging functions have suggested that these pathways might share a wide range of overlapping targets. Published studies have focused on the long-term effects (24–48 h) of FGF and TGF-β inhibition in hESCs, identifying direct and indirect target genes. In this study, we focused on the earliest transcriptome changes occurring between 3 and 9 h after FGF and TGF-β inhibition to identify direct target genes only. Our analysis clearly shows that only a handful of target transcripts are common to both pathways. This is surprising in light of the previous literature, and has implications for models of cell signaling in human pluripotent cells. In addition, we identified STOX2 as a novel primary target of the TGF-β signaling pathway. We show that STOX2 might act as a novel SMAD2/4 cofactor. Taken together, our results provide insights into the effect of cell signaling on the transcription profile of human pluripotent cells

## 1. Introduction

Pluripotent embryonic stem cells (PSCs) harbor great potential for the development of new therapies [1]. While progress is being made to address many risks associated with stem cell transplantations such as oncogenicity, an increasing number of clinical studies and successful applications are paving the way for further expanding treatment strategies (reviewed in [2]). In addition to facilitating clinical applications, understanding of human PSC biology will also be valuable for better characterizing early human development. Human embryonic stem cells (hESCs) are derived from the inner cell mass (ICM) of blastocyst stage embryos and can be maintained in vitro using different culturing protocols. Current approaches aim to target specific signaling pathways by adding ligands. Combined activation of the fibroblast growth factor (FGF) and transforming growth factor β (TGF-β) pathways is sufficient and necessary for hESCs maintenance [3,4].

FGF supplementation is essential for maintaining a pluripotent developmental potential and self-renewal capabilities of human PSCs. FGF cooperates with TGF-β to drive NANOG expression [5,6]. It has also been suggested that FGF2 plays a role in blocking extraembryonic differentiation. FGF receptor inhibition leads to the upregulation of trophectoderm (TE) and primitive endoderm (PE) markers [7,8]. It has been suggested that supplementing the hESC medium with FGF2 can stimulate either feeder cells often used to support hESC growth or the hESCs themselves directly to produce ACTIVIN and IGF2 [9]. Both signaling ligands are beneficial for pluripotency and cell survival. FGF2 has also been suggested to directly inhibit caspase-activated apoptosis via anoikis [10], which is triggered by a lack of cell contact or extra cellular matrix attachment [11]. FGF2 signaling is supporting pluripotency in hESCs in several ways and remains to be fully elucidated [12].

The TGF-β pathway is crucial for early embryogenesis [13] and hESCs [14]. In the context of embryonic stem cells, TGF-β is required for the maintenance of pluripotency [5], and regulates the entry into differentiation [15]. This signaling pathway is spatiotemporally controlled on multiple levels [16,17,18]. The extracellular signal is relayed from the membrane to the nucleus within minutes and effects on the transcriptome can be detected within hours [19]. Ligands induce the formation of a hetero-tetrameric complex of TGF-β-Type I and TGF-β-Type II receptor which phosphorylate receptor SMADs that, together with SMAD4 translocate to the nucleus, bind DNA and regulate transcription [20]. To elicit cell type-specific transcription signatures, the SMAD complexes associate with cell type-specific transcription factors [21]. In hESCs, ACTIVIN and NODAL are the predominant ligands that induce the TGF-β pathway. Long-term inhibition of this pathway in hESCs leads to the exit from pluripotency and commitment to neuroectoderm [22].

Previous studies have investigated the TGF-β [5,15] and FGF [23] pathways in maintenance of pluripotency and differentiation by applying small molecule inhibitors (SMI) to selectively block the pathway followed by transcriptome profiling. However, SMI were applied for 12, 24 or 48 h. The late readout of transcriptional changes makes it likely that secondary effects were observed as morphological changes suggest the onset of differentiation. To identify the primary effects and eliminate to a large extent cascading and crosstalk effects, we performed a high-resolution time course immediately after addition of SMIs for either the TGF-β or FGF pathways. Surprisingly, in-depth bioinformatic analysis revealed only a limited number of co-regulated transcripts after inhibition of the FGF or the TGF-β pathway in hESCs. We attributed this limited transcriptional crosstalk of these pathways to their intrinsic different mode of action. FGF signaling has broad pro-proliferative and survival effects, whereas TGF-β acts directly on the transcriptome by activating SMAD transcription factors.

However, we identify STORKHEAD BOX2 (STOX2) as a novel target of TGF-β in hESCs. Our results show that one mechanism of STOX2 function in hESCs is might be through stabilization of SMAD2 and SMAD4 proteins.

## 2. Materials and Methods

Materials: (antibodies, chemicals, oligonucleotides, vector sequences and medium composition) are listed in Appendix A.

### 2.1. Cell Culture

hESCs (WA01:H1 and WA09:H9) were obtained from WiCell and cultured in E8 medium on Matrigel [24]. 293T cells were cultured according to standard procedures and transfected as described previously [16].

### 2.2. RNA-Sequencing

H1 cells were grown to ~80% confluency on Matrigel in mTeSR medium in Falcon 6-well plates. Medium was replenished 2 h before pathway inhibition. Cells were treated in triplicate with either 10 µM SB431542 for TGFβ inhibition, 10 µM SU5402 for FGF inhibition or an equal amount of DMSO as solvent control. Drug and DMSO treatments lasted 3, 6 and 9 h. Total RNA from treated cells along with untreated control cells from time points 0 h and 9 h was extracted with Trizol according to the manufacturer’s instructions, measured on a Nanodrop and checked for integrity on a 1.5% Agarose gel. An amount of 5 µg of total RNA was handed over to the Functional Genomics Center Zurich, where sample libraries were prepared, polyA-enriched and sequenced on an Illumina HiSeq 2000.

### 2.3. Data Pre-Processing and Quality Control

Initial quality control was performed using FastQC (v0.11.6) followed by low-quality reads filtering and adapters removal by trimmomatic (v0.32) [25]. Reads were then mapped to the human hg38 genome using the STAR aligner (v2.5.0c) [26] allowing a maximum of two mismatches. Read quantification was performed by featureCounts (v1.4.5-p1) [27] using the Ensembl reference (GRCh38.83).

### 2.4. Time Course Differential Expression Analysis

Count data was pre-filtered, requiring each gene to have expression of at least 1 count per million mapped reads (CPM) within each experiment. EdgeR (v3.18.1) [28], ImpulseDE2 (v1.0.0) [29], splineTC (v1.4) [30] and next maSigPro (v1.48.0) [31] were run with standard parameters and candidates were subjected to a p-value filtering of 0.05 and 0.01 for SB and SU experiments, respectively.

### 2.5. Gene Networks

Network data were obtained from STRING [32] using standard parameters and removal of unconnected nodes. Obtained network files and average fold change files for each experiment was loaded into Cytoscape (v3.5.1) [33]. Node color was adjusted to the first time point that had an absolute log fold expression change of 0.5 and 0.7 for SB and SU, respectively. Segmented rings indicating general expression changes were visualized using the circoschart of the enhancedGraphics plugin [34]. The combined network was created by the merge network function of Cytoscape. The common target network was extracted by selecting adjacent nodes of shared targets of SB and SU networks.

### 2.6. Data Visualization

Venn diagrams were created in R (v 3.4.1). Heatmaps and bar-plots were generated in Prism 8.0.

### 2.7. Generation of STOX2 Plasmids for Overexpression

The STOX2A coding sequence was amplified from cDNA by PCR and cloned into expression plasmids by standard techniques. For overexpression of STOX2A (NM_06410.1) in HEK293T cells, the cDNA was cloned into pCMV5-3xFLAG or pCMV5. For overexpression of STOX2A in hESCs, the cDNA was cloned into pBSK-EF1A in frame with eGFP.

### 2.8. Transfection of hESCs

Transfection of hESCs with non-lipid-based reagents is based on [35]. One day prior to transfection, the desired amount of 6-well plates were coated with Matrigel at room temperature for 1 h. Before splitting the cells a 70–80% confluent 6-well plate was pretreated for 1–2 h with regular growth medium plus 10 µM Rock inhibitor. For passaging cultures, cells were rinsed with 1X PBS, and subsequently incubated with 1 mL of Accutase per 6-well plate for 5–10 min until cells detached from the plate and formed single cell suspensions. 10 µM Rock inhibitor Y-27632 was added for enhanced single cell survival. Cells were seeded at a ratio of 1:4–1:6 into the pre-coated wells to generate a monolayer of single cells. On the day of transfection, 3–4 h prior to transfection, the cells were incubated with 1 mL D-PBS for 5 min to increase cell–cell spacing. Afterwards, cells were nutrient and growth factor starved in Opti-MEM with 10 µM Rock inhibitor Y-27632 until transfection for 4–5 h. After starvation, medium was changed to 2.5 mL antibiotics free growth medium per well including 10 µM Rock inhibitor. Transfection mixture was prepared by diluting 2 µg of plasmid DNA in 200 µL OptiMEM in a 1.5 mL Eppendorf tube and adding 6 µL of transfection reagent (Genejuice), vortexing for 5–10 s and incubating for 10 min at room temperature. Transfection mix was then added dropwise to the cells.

### 2.9. siRNA Mediated Knock-Down of STOX2 in hESCs

The transfection protocol was previously described in [36]. The sequences of the siRNAs used are provided in Appendix A.

### 2.10. Overexpression of STOX2 and SMAD2 in HEK293T Cells

Transfections of HEK293T were performed as previously described [16], and the constructs for SMAD2 have previously been published [16].

### 2.11. RNA Isolation and qRT-PCR

Total RNA was isolated using TRIzol according to the manufacturer’s instructions. 1 µg of total RNA was reverse transcribed using the Promega GoScript reverse transcription set and random hexamer primers. Quantitative real-time RT-PCR was performed with gene specific primers (Appendix A) and 2xSYBR green master mix (Kappa) using a Roche 480 light cycler. Relative expression levels of genes of interest compared to GAPDH and HPRT were calculated using the ΔΔCt method.

### 2.12. Statistical Analysis

Statistical analysis was performed using PRISM 8 as indicated.

### 2.13. Chromatin Immunoprecipitation

For Chromatin-Immunoprecipitation, 10 cm dishes of nearly confluent H1 or H9 human ES cells were washed once with 1X PBS, then fixed with 4% Formaldehyde/PBS for 10 min. Crosslinking was quenched by replacing the Formaldehyde solution with 0.125 M Glycine/PBS. Plates were washed with 1X PBS afterwards, scraped down in 5 mL 1X PBS containing 0.05 mL of 100 mM PMSF and centrifuged for 10 min at 720× *g*. At this point, pellets were either stored at −80 °C or used for chromatin shearing. Before shearing, cells were resuspended in 1 mL hypotonic lysis buffer (25 m M HEPES, pH 7.8, 1.5 mM MgCl_2_, 10 mM KCl including protease inhibitors) and incubated for 30 min on ice, followed by 20× douncing. After centrifugation for 5 min at 2500× *g* RMP, 4 °C, the pellet was resuspended in 1 mL sonication buffer (50 mM HEPES pH 7.9, 140 mM NaCl, 1 mM EDTA, 1% Triton X-100, 0.1% Na-Deoxycholate, 0.1% SDS) and transferred to a Covaris Millitube. Chromatin was sheared at 4–8 °C with previously tested conditions to generate DNA-Fragments of about 200–400 bp length: Peak Power: 140, Duty Factor 5.0, Cycles/Burst 200, time 960 s. Sheared chromatin was centrifuged at 16,000× *g* for 10 min. A 50 µL aliquot was de-crosslinked by adding 130 µL H_2_O, 8 µL 5 M NaCl, 10 µL 1 M Tris-HCl pH 8 for 4 h at 65 °C. Followed by RNAseA digest, Proteinase K digest and Phenol-Chloroform extraction, DNA concentration was measured using a Nanodrop.

Chromatin was diluted to 10 µg in 400 µL sonication buffer, 2.5 µg antibody or appropriate IgG control was added per sample and incubated gently rotating for 4 h at 4 °C. 10 µg was saved as input control. 25 µL Protein G Dynabeads were washed once in sonication buffer before being added to the samples and incubated gently rotating for 2 h at 4 °C. Afterwards, beads were washed twice for 5 min with ChIP buffer A, 2x for 5 min with ChIP buffer B, 2x for 5 min with ChIP buffer C, and 1x with TBS, before finally being eluted in 100 µL freshly prepared 50 mM NaHCO3, 50 mM Tris-HCl pH 8, 2 mM EDTA, 1%SDS for 30 min at 37 °C. Supernatant was removed from beads on a magnetic rack and DNA was de-crosslinked from proteins by adding 4 µL 5 M NaCl per sample and incubated for at least 4 h at 65 °C. After de-crosslinking, residual RNA was digested with 1 µL RNAseA (10 mg/mL) at 37 °C for 20 min. The remaining protein was digested with 2 µL Proteinase K (10 mg/mL), 5 µL of 1 M Tris/pH 6.8, and 1µL of 500 mM EDTA for 4 h at 37 °C. DNA was then phenol-chloroform purified and eluted in 50 µL elution buffer. 2 µL was used per qRT-PCR reaction.

### 2.14. DNA-Pulldown

400–600 bp promoter or enhancer sequences were determined using UCSC genome browser displaying relevant histone marks or TF factor binding. Promoter and enhancer sequences were then amplified from wildtype genomic DNA and cloned into pBluescript vector for further amplification. Biotinylating PCR was carried out with a biotinylated T7 and an unbiotinylated T3 primer. Multiple PCR reactions were pooled, precipitated and purified with a Qiagen PCR purification kit. For DNA pulldown, 60 µL streptavidin-agarose bead suspension was washed twice in PBS and then incubated with 2 µg biotinylated PCR product in Biotin-Streptavidin Buffer (10 mM Tris pH 7.5, 1 mM EDTA, 1 M NaCl, 0.003% NP40) for 1 h at room temperature. Beads were then blocked with 50 µM Biotin in Biotin-Streptavidin buffer for 15 min at room temperature followed by two washes in 1 mL Biotin-Streptavidin Buffer each. Reaction solution containing 150–250 µg protein lysate was combined with the coupled beads and incubated over night at 4 °C gently rocking. The next day, the beads were washed five times with 1 mL Binding buffer each (20 mM HEPES pH 7.5, 2.5 mM KCl, 1 mM DTT, 20% Glycerol, 0.01% NP40). After the last wash, all supernatant was removed, and the beads were resuspended in 50 µL 2x Laemmli buffer.

### 2.15. Protein Isolation, Immunoprecipitation and Immunoblot Analysis

Total protein extracts were obtained from cells lysed in TNTE buffer (50 mM Tris/HCl pH 7.6, 150 mM NaCl, 0.5% TritonX-100, 1 mM EDTA) containing protease and phosphatase inhibitors and protein concentration determined by BCA protein determination kit (Pierce). Immunoprecipitations were performed with antibodies against FLAG, HA or STOX2 and protein dynabeads. For immunoblot analysis, proteins were separated by SDS-PAGE, transferred to nitrocellulose membranes and blocked in 5% non-fat dry milk in TBS-T (10 mM Tris/HCl pH 7.5, 150 mM NaCl, 0.1% Tween 20). The antibodies were applied in the blocking solution at the indicated dilutions (Appendix A) at 4 °C overnight. The blots were washed three times with TBS-T, incubated with HRP coupled antibodies (Jackson Immuno Labs) and developed on a BioRad imager using ECL (Pierce). Before re-exposing, the blots were incubated with stripping buffer (1 M Glycine pH 2.5, 0.5% SDS) for 1 h and extensively washed.

### 2.16. Differentiation of hESCs

Mesendoderm: Cells were grown to 50% confluency and before initiating differentiation, cultures were given a brief wash with 1X PBS. Differentiation was carried out in RPMI supplemented with 2 mM glutamine, 100 µg/mL penicillin/streptomycin. The FBS concentration was gradually increased during the differentiation time course from 0% for the first 24 h, 0.2% for the second 24 h and 2.0% for the subsequent days of differentiation with continuous exposure to 25 ng/mL Activin A supplemented with 1X B27 [7].

Neuroectoderm: Neural differentiation was carried out as described by [22]. Cells were grown to near confluency and before initiating differentiation, cultures were given a brief wash with 1X PBS. To start neural induction, the medium was changed to Neural Induction (NI) medium, containing inhibitors for TGF-β and BMP signaling. The medium was changed every day for 3 days.

### 2.17. Data Availability

All raw RNA-seq data and raw read counts can be accessed at the Gene Expression Omnibus database under the accession number GSE123944.

### 2.18. Ethical Approval

Experiments with hESCs were approved by the cantonal (Zurich) and federal authorities (Switzerland) (ref: R-FP-S-1-0008-000).

## 3. Results

To identify direct and common targets of TGF-β and FGF signaling pathways, we treated hESCs with SMI (SU5402 (SU) or SB4315432 (SB), and Dimethyl sulfoxide (DMSO) as control), for 0, 3, 6 and 9 h. In addition, we harvested untreated hESCs at 0 h and 9 h to account for potential solvent effects on the transcriptome. Subsequently, we extracted total RNA from the cultures (Figure 1A) and prepared polyA RNA libraries for high-throughput sequencing.

### 3.1. Identification of TGF-β and FGF Direct Target Genes

Inhibition of signaling pathways by blocking the corresponding cell surface receptors by SMI prevents phosphorylation and activation of adapter proteins and consequently aborts downstream signal propagation. This in turn induces changes in expression of downstream targets. To verify the inhibitory effects, we first investigated whether known direct targets of the FGF and TGF-β pathway were downregulated in the SU and SB treated time course. The TGF-β targets SMAD7, NODAL, LEFTY1, LEFTY2, and NANOG were strongly downregulated after SB treatment (Figure 1B). Likewise, DUSP6 and DUSP5, known FGF targets, were also downregulated upon SU treatment (Figure 1B). Notably, pluripotency markers expression was not affected at early time-points, with the exception of NANOG, which is a direct TGF-β and FGF target (Figure 1B). This observation indicates that we were able to capture transcriptional changes induced by blocking both the FGF and TGF-β signaling pathways without changing the pluripotent identity of the hESCs. Thus, our dataset facilitates the dissection of immediate effects and direct targets for both pathways.

#### 3.1.1. Differential Time Course Gene Expression Analysis

Nowadays, the analysis of RNA-seq time series is still challenging, and a gold standard method remains to be established. Therefore, we evaluated several strategies by conducting a simulation-based benchmarking study [37]. The performance on simulated data sets was used to select bioinformatics tools for differential expression (DE) analysis of our experimental data. Three tools were included in this analysis: splineTC [30], maSigPro [31], and impulseDE2 [29]. In addition, we performed standard pairwise comparisons using edgeR [28] (Appendix A). DEGs of tools were intersected for TGF-β (Figure 1C, left) and FGF signaling (Figure 1C, right). To avoid biases from stringent filtering, the minimal log2 fold change (LFC) of 0.5 was chosen for SB (TGF-β). For FGF signaling (SU) we chose the threshold of 0.7 log2 fold change based on NANOG, a well-known target gene of this pathway in hESCs [38]. Repeated overlap of LFC candidates from different prediction tools revealed increased number of shared candidates with the exception of edgeR and ImpulseDE2 in SB and SU, respectively, which remained ~50% exclusive (Figure 1C, Appendix A). In addition to the LFC threshold, we decided to use candidates identified by at least three tools, obtaining 74 and 275 candidates for SB and SU, respectively (Figure 1C, highlighted in bold).

#### 3.1.2. Gene Network Visualization

Due to the cascade character of signaling pathways, the delay caused by the propagation of activation through the components of the signaling cascade results in an ordered series of transcription changes over time. We expected that orchestrated DE candidates should be observed when creating a gene network and adding temporal information of DE. To this end, we extracted known protein–protein interactions of our candidates from the STRING database [32] and removed orphan nodes. While the resulting SB network was very concise consisting mainly of known TGF-β pathway components (Appendix A), the SU network encompassed a wider range of genes (Appendix A). Temporal succession has been visualized by using the first time point of DE as node color and a segmented ring within to indicate general up/down regulation for each time point.

We observed temporally correlated expression changes in both experiments. Upon SB treatment, SMAD7 is depleted and cannot longer inhibit TGF-β [39] or epithelial–mesenchymal transition (EMT) [40], leading to the upregulation of both EMT actor SERPINE1 [41] and EMT suppressor HMOX1 [42]. To illustrate another example in SU, we highlight the early upregulation of IRF2BP1, which has been reported to enhance the repressive function of JDP2. JDP2 associates with FGF/JNK factor JUND that is binding ATF2 or AP-1 motifs and subsequently suppressing targets such as PLAU [43], thereby abolishing PLAU functions in extracellular matrix remodeling and adhesion, both processes associated with differentiation of hESCs [43].

Taken together, the integration of interaction data made it possible to create networks and hierarchies of interactions, which were further refined by adding expression changes from our time-course RNA-seq data. Extraction of sequential interactions further allowed the identification of related biological processes. FGF and TGF-β pathways are both involved in remodeling the cell towards a new cell specification, the first process by disrupting negative feedback loops necessary for the maintenance of pluripotency, and the second process by activating repressors of signaling.

FGF and TGF-β pathway crosstalk was assessed by network merging and filtering for direct interactions with candidates identified in both drug inhibition experiments (Figure 1D). By reducing the size of the network and its complexity, the candidates mediating crosstalk between TGF-β and FGF signaling pathways were selected. Chosen targets (Figure 1E) encompassed transcription factors NANOG, ID1 and ELF3, hedgehog receptor PTCH1, ERK/ELK1 regulator IL17RD, cell migration or ECM associated genes NTN1 and SERPINE1, calcium signaling associated P2RY1 and insulin signaling/TP53 effector IGFBP3. Interestingly, 16 of the 23 common differentially expressed genes were similarly regulated by both pathways suggesting a synergistic regulation by FGF and TGF-β (e.g., NANOG). In contrast, ID1, a factor associated with pluripotency and repression of differentiation, was downregulated upon TGF-β inhibition and induced by suppression of FGF signaling. PTCH1 was upregulated after inhibition of both TGF-β and FGF, and might have been further antagonizing WNT signaling, thereby preventing differentiation [44]. Taken together, our data suggest a complex regulatory signaling network that remains to be fully characterized in future studies.

#### 3.1.3. Identification of Novel Direct TGF-β Targets

Among the 74 candidates regulated by TGF-β signaling, we found previously identified TGF-β target genes including: NODAL (log2 fold change 1.01 (3 h), 3.02 (6 h), and 2.92 (9 h)), SMAD7 (log2 fold change 0.84 (3 h), 1.68 (6 h), and 1.96 (9 h)), and NANOG (log2 fold change 0.58 (3 h), 1.24 (6 h), and 1.54 (9 h)) (Figure 2A, Appendix A overlap list). Among the 275 DEG identified after suppression of FGF signaling were the known FGF target genes EGR3 (log2 fold change 3.35 (3 h), 2.24 (6 h), and 5.22 (9 h)), EGR1 (log2 fold change 1.88 (3 h), 1.53 (6 h), and 2.67 (9 h)), DUSP5 (log2 fold change 1.72 (3 h), 1.33 (6 h), and 1.61 (9 h)), DUSP6 (log2 fold change 1.71 (3 h), 1.52 (6 h), and 2.34 (9 h)) (Figure 2B, Appendix A overlap list). Interestingly, we found Storkhead Box2 (STOX2), which has not been associated with TGF-β signaling in hESCs before, in the top seven downregulated transcripts (log2 fold change 0.61 (3 h), 1.75 (6 h), and 2.06 (9 h)) (Figure 2A). STOX2 is a member of the WINGED HELIX DOMAIN-containing proteins, which consists in mammals of two paralogues, STOX1 and STOX2. Orthologue proteins are present in all vertebrates, as well as in model organisms such as Drosophila melanogaster and Caenorhabditis elegans (PANTHER FAMILY; PTHR22437). STOX1 has been shown to act as a transcription factor binding DNA via its WINGED HELIX DOMAIN [45] and is indirectly suppressed by NODAL signaling in extravillous trophoblasts [46]. In contrast to STOX2, STOX1 seems not to be regulated by TGF-β signaling in our analysis of hESCs, indicating that STOX 1 and STOX2 share some similarities but might have distinct mode of action. STOX2 has been associated with pre-eclampsia [47] and oral squamous cell carcinomas [48]. Interestingly, the Drosophila and the C. elegans orthologues ko and ham-1 have been implicated in asymmetric division of neuronal precursor cells [49].

To validate the transcriptome analysis, we performed qRT-PCR and showed that STOX2 is regulated by TGF-β signaling in hESCs (Figure 2C). We used as control genes NANOG, LEFTY1 and LEFTY2. We also confirmed regulation of STOX2 by TGF-β at the protein level, although with a slightly slower kinetic. To confirm the effect of SB, we also investigated loss of phospho-SMAD2 using SMAD2 and phospho-SMAD2 (pSMAD2) antibodies (Figure 2D).

To determine whether SMAD proteins directly regulate STOX2 expression, we investigated published Chromatin immunoprecipitation sequencing (ChIP-seq) data sets [21,50] for potential SMAD2, as well as NANOG, OCT4 and CTNNB1 binding sites around the STOX2 gene locus. We identified two potential regulatory regions. One site is located directly upstream of the translational start site and a second intronic site between exon 1 and exon 2 (Figure 3A, upper part). ChIP-qPCR experiments established that both sites were occupied by SMAD2 in a TGF-β dependent manner (Figure 3A, lower part). To ensure specificity of the SMAD2 ChIP, we investigated regions that showed no apparent SMAD2 binding in published ChIP-seq data [21]. These control regions did not show an enrichment of SMAD2 ChIP signal compared to IgG control precipitation (Figure 3A). To answer the question whether other transcription factors are involved in regulating STOX2 in hESCs, we performed DNA pull-down using biotinylated DNA fragments coding for the identified regulons as described previously [16]. These experiments showed that the STOX2 regulatory regions are not only bound by SMAD2 in a TGF-β dependent manner, but also by the core pluripotency factors OCT4 and NANOG, as well as CTNNB1, the effector of WNT signaling (Figure 3B). Overall, our results integrate STOX2 as a TGF-β and potentially WNT regulated transcript into the pluripotency network of hESCs.

#### 3.1.4. STOX2 Knock-Down Leads to Decreased Levels of SMAD2 and SMAD4

Our previous results established that STOX2 is directly regulated by TGF-β signaling in hESCs. We used a loss of function approach to elucidate the molecular functions of STOX2 in hESCs by RNA interference [36]. siRNAs-mediated depletion of STOX2 led to a reduction in the mRNA expression levels of classical TGF-β targets such as LEFTY1 and NODAL. We did not observe a reduction of SMAD2 and SMAD4 mRNA levels (Figure 4A). However, we observed a loss of SMAD2 and SMAD4 proteins, as well as SMAD2 phosphorylation (Figure 4B).

#### 3.1.5. Overexpression of STOX2 Leads to an Increase in TGF-β Signaling Activity

To add further support to our hypothesis that STOX2 is a novel component of the TGF-β pathway in hESCS, we overexpressed STOX2 in these cells. In STOX2 overexpressing cells, we observed an increase of LEFTY2 by 20%, and near 2-fold increase of NODAL at both mRNA and protein levels (Figure 4C,D). Concomitant with the increase of STOX2 protein levels, SMAD2 protein became slightly more abundant compared to non-transfected cells, whereas we did not detect changes in SMAD2 mRNA (Figure 4C,D).

Taken together, loss of STOX2 in hESCs leads to a reduction of SMAD2 and SMAD4 proteins that is independent of transcription and results in reduced expression of TGF-β target genes.

#### 3.1.6. STOX2 Interacts with SMAD2 in Whole Lysate Extract

Feedback loops in the TGF-β superfamily often involve direct physical interactions with the target genes and components of the pathway itself. For example, interaction of SMAD7 and its target SNON has been shown [51]. STOX2 has a storkhead box winged helix domain that was suggested to confer DNA binding activity [47]. Therefore, we speculated that STOX2 might physically interact with the transcriptional components of the TGF-β pathway, SMAD2/3 and/or SMAD4. To test this hypothesis, we transfected 239T cells with STOX2 and SMAD2 and performed immunoprecipitations followed by immunoblot analysis. Immunoprecipitation using FLAG (Figure 5A) or HA antibodies (Figure 5B) showed STOX2 precipitated with Flag-SMAD2 and vice versa. Ectopic expression of epitope tagged proteins for physical interaction studies can yield false positive results. For further validation at the endogenous level, we performed immunoprecipitation of STOX2 from cell lysates of hESCs either treated with DMSO (solvent control) or SB (inhibition of TGF-β signaling). Endogenous SMAD2 was precipitated efficiently by STOX2 in the presence of TGF-β signaling (Figure 5C). After inhibiting the pathway, STOX2 no longer interacted with SMAD2. In summary, these experiments show, that STOX2 might form a complex with SMAD2 in an TGF-β dependent manner. However, our experiments cannot determine whether the observed interaction is direct or mediated via other proteins. Therefore, further experiments should be performed to clarify the nature of the interaction in the future.

#### 3.1.7. STOX2 Is Differentially Expressed during hESCs Cell Fate Specification

Signaling by TGF-β in combination with FGF2 is crucial for the maintenance of pluripotency. Combined loss of TGF-β and BMP signaling leads to specification of neuroectoderm [22]. Conversely, increased TGF-β signaling in combination with WNT induces mesendoderm specification [7]. We observed that induction of mesendodermal differentiation leads to a sharp increase of STOX2 mRNA expression after 24 h. This result is consistent with our previous observation that STOX2 is a target of the TGF-β and potentially WNT signaling pathways (Figure 3). Subsequently, STOX2 levels declined over a 48-h time period and reached a level comparable to that of hESCs (Figure 6A). To confirm the efficiency of the mesendoderm specification, we monitored the mRNA expression levels of OCT4, EOMES and GSC (Figure 6A). Therefore, STOX2 is transiently activated at the beginning of mesodermal differentiation of hESCs.

Neurectodermal differentiation of hESCs was accompanied by an initial reduction of STOX2 mRNA. This observation is to be expected, as the differentiation protocol includes the inhibition of TGF-β signaling. However, after 72 h of differentiation, we observed a strong increase of STOX2 expression (Figure 6B) concomitant with induction of the neuroectoderm markers SOX1 and PAX6, and loss of OCT4 expression (Figure 6B). Immunoblot analysis of STOX2 during neurectoderm differentiation showed that STOX2 protein levels followed the mRNA levels with a slight delay (Figure 6C).

During gastrulation, mesendoderm cells arise from epiblast cells, that migrated through the primitive streak, a structure induced by TGF-β, BMP and WNT signaling [52]. Ectoderm-derived tissues emerge from the part of the epiblast that did not enter the primitive streak and therefore were not exposed to signaling from TGF-β, BMP or WNT pathways [52]. Our results show that STOX2 is downstream of TGF-β in the pluripotent state and at the onset of differentiation towards mesendoderm. This suggests that STOX2 might play a role in pluripotency and induction of the primitive streak dependent on TGF-β and WNT. In ectoderm specification, using a protocol that relies on blocking TGF-β and BMP signals, we observed an initial downregulation of STOX2 consistent with our hypothesis that this gene is directly regulated by TGF-β signaling. At later stages of ectoderm specification, however, STOX2 mRNA increases independent of TGF-β, suggesting an alternate mode of regulation. WNT and FGF have been implicated in specification of the neural crest linage that arises between the neural plate and the non-neural ectoderm [53]. Taken together, our findings indicate that STOX2 might be regulated by either TGF-β and WNT and potentially lineage specific TF. Therefore, the context dependent expression of STOX2 remains to be investigated in future studies.

## 4. Discussion

In this study, we aimed to dissect the roles of TGF-β and FGF signaling in hESCs [3] by blocking signal transduction at the corresponding receptor level and investigating early consequences. The effects of signal inhibition on pluripotency and differentiation were further investigated by lineage specific gene expression profiles. Previous studies have used transcription profiling, but were limited by investigating late timepoints, where secondary effects predominate. In contrast, we measured transcriptional changes that occur immediately after signal inhibition. Thus, our strategy avoids changes of cell identity through cell division or secondary and tertiary effects of interconnected signaling cascades. One challenge of our analysis arises from the initially modest changes in transcript abundance and hence detection thresholds for gene expression changes art very early timepoints. To overcome this problem, we implemented a data analysis strategy that is based on performance recommendations of a previously conducted simulation-based benchmarking study for time course RNA-seq data [37]. Our strategy takes advantage of several tools, including ImpulseDE2, next maSigPro and splineTC. In addition, we perform pairwise comparison using edgeR. Combining predictions from several tools that emphasize different aspects for detecting gene expression changes, we were able to dramatically improve the sensitivity for candidate identification. Candidates of each tool were intersected and subsequently ranked by overlap rates between the individual tools and fold change expression compared to control (Figure 1). Surprisingly, SB drug treatment had less pronounced effects on the transcriptome, indicated by the number of DEGs identified compare to SU drug treatment. Stronger effects would have been more intuitive, as TGF-β directly feeds into the core pluripotency network by activating NANOG [54], whereas FGF signaling is thought to sustain this regulation [3,55]. TGF-β signaling acts directly on promoters via SMADs, whereas FGF signaling involves the phosphorylation of downstream targets and has wider effects in the cell. Repression of feedback loop inhibitors such as DUSP6 [56], SPRY4 [57] and IL17RD [58], which are also clustered in our obtained networks (Appendix A), might potentially limit the effects of loss of TGF-β signaling.

To filter out noisy signal, a minimal fold change filter was applied, whereby an experiment-specific cutoff had to be specified. The choice of data transformation was guided to maintain the sensitivity based on previously known target genes. Standard thresholds are usually set to a 2-fold change, though considering the short treatment and nature of signal modulation effects, we set the threshold to 1.4 for SB. In SU experiments, a wider variety of expression changes were observed. To this end, we used NANOG levels as the threshold, as they represented the best positive control, for identifying direct targets. Thus, pointing out the importance of positive controls to ease candidate filtering and to set thresholds based on the literature. Final candidate lists were investigated for functional groups and hierarchical structures. Extraction of interaction information from the STRING database allowed the creation of gene networks (Figure 1D, Appendix A). The advantages of the STRING database include the combination of several data type sources, including text-mining approaches to connect candidates. While STRING offers the possibility of extending the network by adding intermediate nodes, the complexity and size of the networks increases exponentially. A possible workaround would be to only extract intermediate nodes for singleton nodes, e.g., as supported by MoCha [59]. While the obtained networks recreated functional groups and hierarchies, only ~50% of candidates were incorporated (Appendix A). Furthermore, additional information can be plotted into gene network maps. Instead of using the connectivity as the node size, the expression fold change or the maximum expression change might be more informative for extracting candidates of interest. Integration of external data such as ChIP-seq binding or regulatory relationships will further increase functional understanding of the underlying processes. Network merging (Figure 1D) presents an easy way to visualize common direct candidates for both pathways and their intermediate signaling environment and cooperation partners. Additional information could be integrated, such as the first time point of differential expression for each experiment as two split nodes, though any additional information would simply increase the complexity and greatly reduce the readability of visualizations. Possible alternatives would be to cluster functional groups such as genesets. This could also increase connectivity with former uncategorized candidates by assessing interplay on a set instead of gene level.

With our approach, we intended to test the hypothesis, that the FGF and TGF-β pathway interact to regulate target genes responsible for maintaining the pluripotent state. Surprisingly, the transcriptomics approach yielded only a handful of target genes regulated by both pathways. Among them where well know factors such as NANOG and ID1 consistent with previous studies, and PTCH1, SERPINE1, TGFBP3, ELF3, IL17RD, NTN1 and P2RY1. The limited number of common targets might be explained by the fact that these two pathways have distinct molecular modes of action. TGF-β signaling acts mostly by directly activating SMAD transcription factors to regulate distinct transcriptional programs. FGF, on the other hand, acts more broadly via activation of cascades through RAS/MAPK, PI3K/AKT, JAK/STAT and PLCgamma/PKC. Furthermore, FGF regulates not only transcription, but also cell proliferation, metabolism and survival. Taking this into account, we reasoned that by only looking at changes in the transcriptome, we might not capture the entirety of the functions of these pathways in hESCs. In the future, it will be interesting to apply additional approaches such as total and phospho-proteome as well as metabolome profiling to complement our current data set.

In our computational analysis, we noted that among the TGF-β targets was STOX2, a factor that has not been implicated in TGF-β regulation before. Therefore, we focused our efforts on this understudied factor. Here, we propose STOX2 to be a novel signaling co-factor modulating the TGF-β pathway in hESCs. Upon exit from pluripotency, STOX2 is regulated differentially depending on the cell fate chosen. Cells resembling the primitive streak in the gastrulating embryo receive high amounts of TGF-β signals and show an immediate increase of STOX2 mRNA levels, followed by a sharp decline. This confirms that STOX2 mRNA expression is indeed directly controlled by TGF-β in hESCs. In contrast, cell populations that will form the future ectoderm receive no TGF-β signaling, and therefore show decreased STOX2 expression. Concomitant with the appearance of markers of neurectoderm (PAX6, SOX1), STOX2 mRNA levels rise. These results suggest that STOX2 is regulated by different signaling pathways depending on the developmental context, and possibly different lineage transcription factors. Among the possible candidates taking over STOX2 regulation at later stages of neurectoderm development are WNT, FGF and SHH signaling [22,53]. In particular, WNT could be a promising candidate, given our results that the WNT downstream effector CTNB is bound to regulons in the STOX2 gene (Figure 2C).

Our study focuses on the molecular functions of STOX2 in the context of pluripotency, and downstream of TGF-β signaling. Interestingly, in the absence of TGF-β, we observed increased binding of CTNNB1, as well as NANOG to the enhancer region located between exon 1 and 2, suggesting a complex regulatory mechanism in pluripotency. Additional studies will be of interest for determining the exact molecular mechanism and further exploring the potential crosstalk between WNT/CTNNB1, TGF-β and the core pluripotency factors OCT4 and NANOG in the regulation of STOX2.

An important insight from our study is that STOX2 not only binds SMAD2 but also is required for SMAD2 protein stability. Suppression of STOX2 levels in hESCs leads to reduced SMAD2/4 protein levels and reduced expression of TGF-β target genes (Figure 4B). These findings show that STOX2 is not only a transcriptional target, but also an intrinsic component of TGF-β signaling in hESCs (Figure 5). Loss of STOX2 in hESCs leads to a destabilization of SMAD2/4 protein levels independently of transcription. This leads us to speculate that STOX2 might interact with SMAD proteins. Indeed, we detected strong interactions between STOX2 and SMAD2 (Figure 5). SMAD protein turnover is tightly controlled by HECT-E3 ubiquitin ligases [20] and interaction with STOX2 might protect SMAD2/4 from proteasomal degradation. In contrast, STOX2 has a potential DNA binding domain, and the fact that it interacts with SMAD2 tempts speculation that it acts as a transcriptional co-factor. However, from our data, it is not clear whether STOX2 acts transcriptionally or solely as factor providing stability to the SMAD proteins. Therefore, future studies are needed to further dissect the functions of STOX2 at the molecular, as well at the functional level to define the exact mode of action in the TGF-β pathway.

In summary, our study identified STOX2 as a new target and component of TGF-β signaling in hESCs modulating the dynamics of SMAD2/4 transcriptional activity by providing protein stability. In future studies, it will be interesting to see whether this novel component of the TGF- β pathway is also present later in development and plays a role in tumor biology.

## Figures and Tables

**Figure 1 biology-09-00470-f001:**
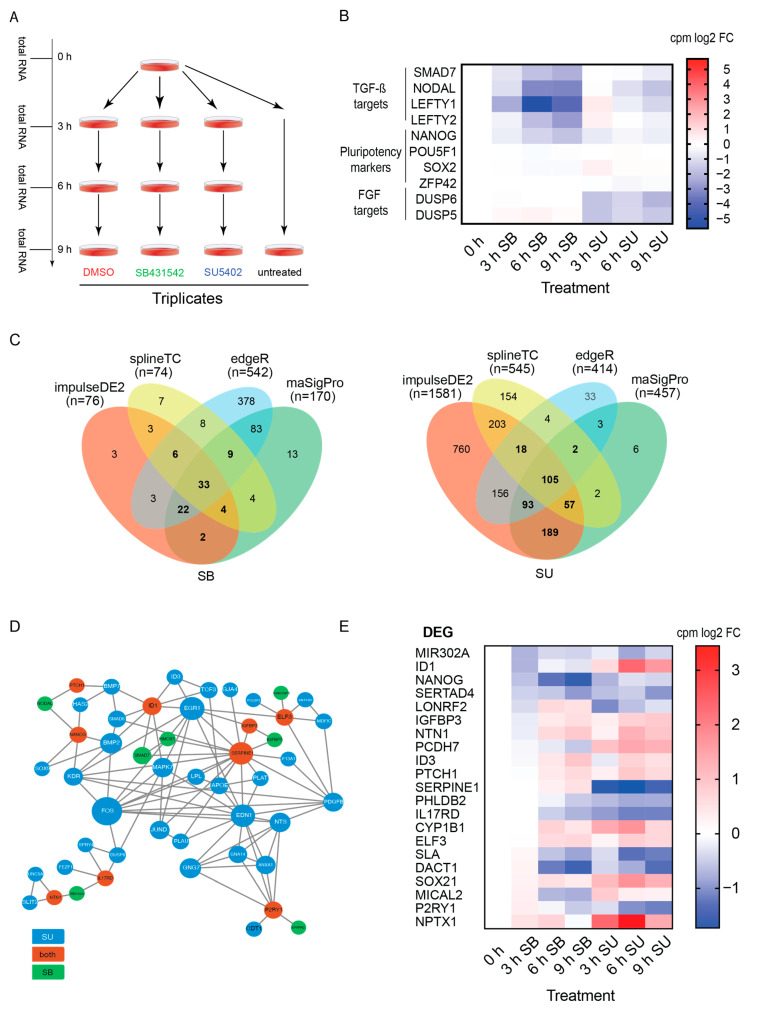
Time course analysis of transcriptional changes upon inhibition of TGF-β and FGF signaling pathways. (**A**) Experimental design of time course RNA-sequencing. (**B**) Heatmap of log2 fold change (compared to DMSO) counts per million expression of know TGF-β and FGF signaling targets compared to pluripotency markers. (**C**) Adjusted Venn diagram for 0.5 and 0.7 LFC filtered candidates of SB and SU, respectively, showing an overall decreased number of tool specific candidates. (**D**) Crosstalk network was generated by merging SB (green) and SU (blue) networks (Appendix A) and filtering for nodes directly connected to candidates identified in both experiments (red). (**E**) Heatmap of log2 fold change (compared to DMSO) counts per million expression of target genes common to the FGF and TGF-β pathways (sorted for maximal downregulation at 3 h of TGF-β inhibition).

**Figure 2 biology-09-00470-f002:**
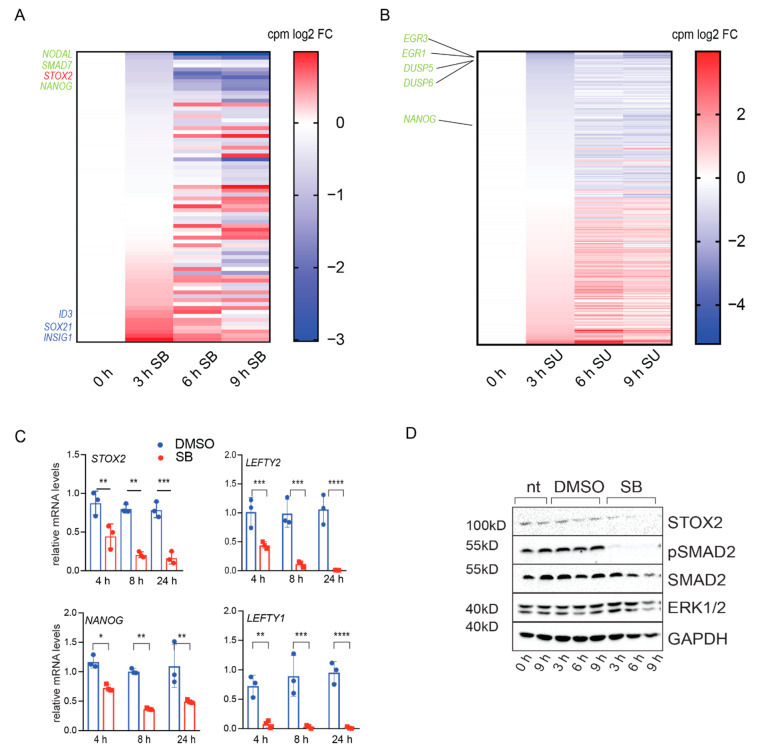
Identification of STOX2 as a novel TGF-β target. (**A**) Heatmap of log2 fold change counts per million (compared to DMSO) of SB regulated target genes. (**B**) Heatmap of log2 fold change counts per million (compared to DMSO) of SU regulated target genes. (**C**) mRNA expression levels of STOX2, NANOG, LEFTY1 and 2 in hESCs (H1) were determined by RT-qPCR after inhibition of TGF-β signaling by 10 μM SB43152 for 4, 8, and 24 h. Treating cells with DMSO served as control. mRNA levels were calculated relative to GAPDH and HPRT using the ΔΔCt method. (N = 3, average ± SD, *p*-value * *p* < 0.05, ** *p* < 0.01, *** *p* < 0.001, **** *p* < 0.0001, ordinary 2-Way ANOVA, Sidak’s multiple comparisons). (**D**) Immunoblot showing STOX2 protein levels in hESCs after 3, 6 and 9 h of TGF-β inhibition by SB (10 μM). Untreated and DMSO treated cells served as controls. Efficient inhibition of TGF-β signaling was monitored by pSMAD2 levels. Equal loading was confirmed with antibodies against SMAD2, ERK1/2 and GAPDH. Representative images of three independent experiments are shown.

**Figure 3 biology-09-00470-f003:**
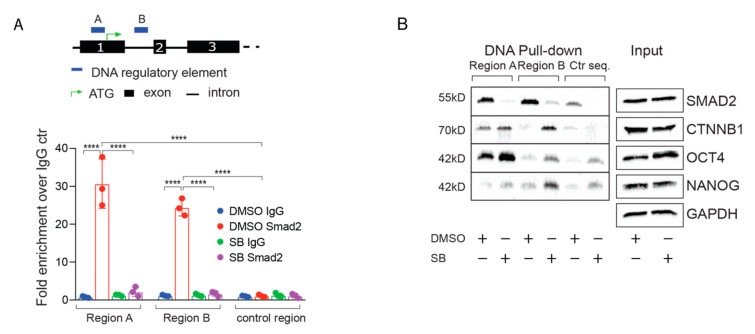
STOX2 is a direct target of TGF-β signaling. (**A**) Cartoon to illustrate the SMAD2 binding regions on the STOX2 gene (Upper panel). SMAD2 binds in an TGF-β dependent manner to DNA regulatory elements in the STOX*2* gene. ChIP-qPCR experiments showing fold enrichment of SMAD2 binding compared to IgG controls in region A, B and an upstream control region in the presence (DMSO) or absence (SB) of TGF-β signaling (illustrated below the bar graph). (N = 3, means ± SD, *p*-value **** *p* < 0.0001, 2-way ANOVA, Tukey’s multiple comparisons). (**B**) DNA pull-down assay with biotinylated probes coding for the two STOX2 regulons identified and a control sequence. hESCs (H1) were treated with 10 μM SB43152 for 2 h, total protein lysates obtained and incubated with the biotinylated DNA segments. After washing, the captured proteins were analyzed by immunoblot using antibodies as indicated. Equal loading was ensured by analyzing aliquots from the total protein lysates before pull-down by immunoblot. Representative blots of three independent experiments are shown.

**Figure 4 biology-09-00470-f004:**
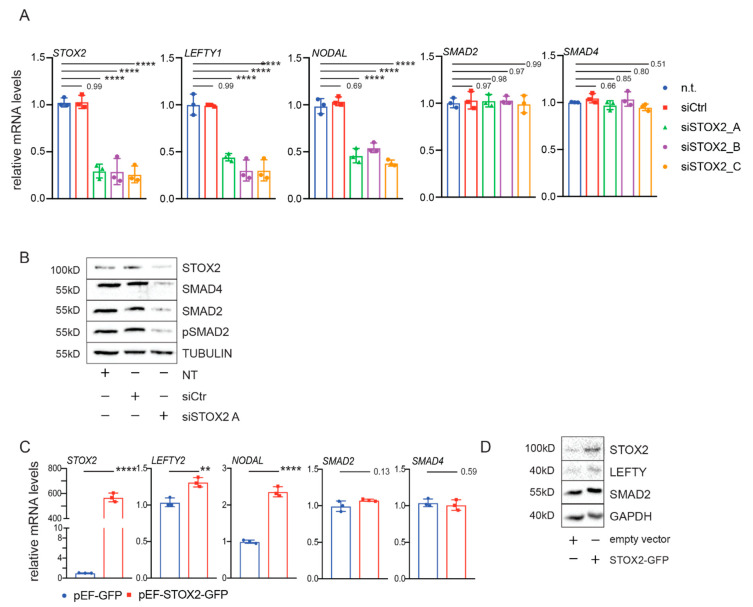
STOX2 misregulation impacts the expression of TGF-β target genes. (**A**) siRNA-mediated knock-down of STOX2 in hESCs (H1). H1 cells were transfected with siRNA targeting STOX2; sictr and mock treated cells served as control. After 48 h, pellets of cells were collected and mRNA expression levels of STOX2, LEFTY1, NODAL, SMAD2 and SMAD4 were determined. Relative mRNA levels to GAPDH and HPRT were calculated using the ΔΔCt method. (N = 3, average ± SD, *p*-value **** < 0.0001, ordinary 2-Way ANOVA, Dunnett’s multiple comparisons). (**B**) Immunoblot analysis with antibodies against STOX2, SMAD4, SMAD2 and pSMAD2 was performed in hESCs (H1) treated as in (**A**). Staining of the membranes with TUBULIN antibody ensured equal loading. Representative images of three independent experiments are shown. (**C**) Overexpression of STOX2-GFP in hESCs (H1). Stable cell lines over expressing STOX2 were generated, and mRNA expression levels of STOX2, LEFTY2, NODAL, SMAD2, and SMAD4 in hESCs stably overexpressing STOX2-GFP were determined by RT-qPCR and relative mRNA to GAPDH and HPRT levels calculated using the ΔΔCt method. (N = 3, average ± SD, *p*-value ** < 0.01, **** < 0.0001 unpaired t-test, two tailed). (**D**) Immunoblot with antibodies directed against STOX2, LEFTY1 + 2 and SMAD2 in hESCs (H1) overexpressing STOX2-GFP as in (**C**). Probing the membranes with antibodies against GAPDH ensured equal loading. Representative images of three independent experiments are shown.

**Figure 5 biology-09-00470-f005:**
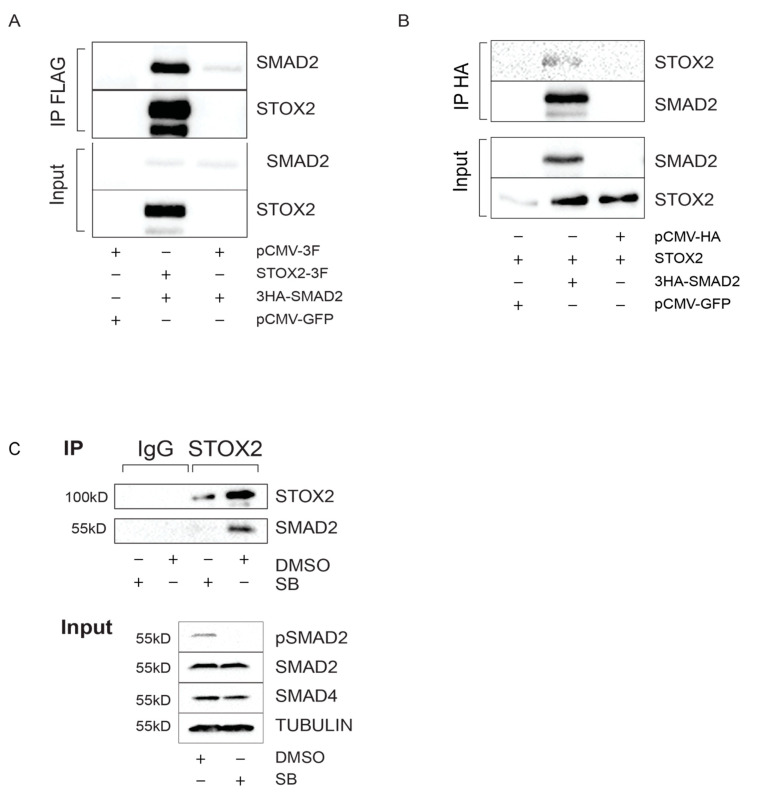
STOX2 interacts with SMAD2 in whole cell extract. (**A**) 293T were transfected as indicated and STOX2-FLAG immunoprecipitated. Subsequent immunoblots revealed that overexpressed SMAD2 interacts with STOX2 (representative images are shown, N = 3). (**B**) 293T were transfected as indicated and SMAD2-HA immunoprecipitated. Subsequent immunoblots revealed that overexpressed SMAD2 interacts with STOX2 (representative images are shown, N = 3). (**C**) Interaction of STOX2 and SMAD2 depends on TGF-β signaling in hESCs. 2 mg of total proteins extracted from hESCs treated for 4 h with DMSO or SB, were incubated with 2 μg of STOX2 or IgG control antibodies. Immunoblot analysis with antibodies against STOX2 and SMAD2 revealed that these proteins from a complex.

**Figure 6 biology-09-00470-f006:**
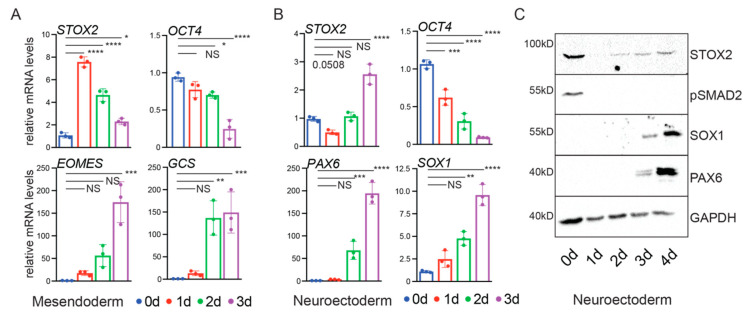
STOX2 is differentially expressed upon cell fate specification of hESCs. Relative STOX2 mRNA levels during mesendoderm (**A**) and neurectoderm (**B**) specification. Indicative transcripts for mesendoderm (EOMES) and (Gooscoid, GCS) (**A**) or neurectoderm (PAX6) and (SOX1) (**B**) were used as readout for effectiveness of the differentiation process. Relative expression levels were calculated by the ΔΔCt method relative to GAPDH and HPRT (N = 3, average ± SD, *p*-value * *p* < 0.05, ** *p* < 0.01, *** *p* < 0.001, **** *p* < 0.0001, ordinary 1-Way ANOVA, Dunnett’s multiple comparisons). (**C**) Immunoblot analysis of STOX2 protein levels during differentiation of hESCs towards neurectoderm. Protein levels of PAX6 and SOX1 are used to indicate the success of the differentiation and levels of pSMAD2 to monitor the block of TGF-β signaling. Equal loading was ensured by staining the membranes with antibodies against GAPDH. Representative images for three independent experiments are shown.

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
