# Peer review of "Inhibition of FGF and TGF-β Pathways in hESCs Identify STOX2 as a Novel SMAD2/4 Cofactor"

_biology, 2020, doi:10.3390/biology9120470_

Round 1

Reviewer 1 Report

The authors have addressed all my concerns and I consider the manuscript now in good shape for publication.

I only notice that Figure 2D legend is missing

Reviewer 2 Report

My concerns have all been addressed.

This manuscript is a resubmission of an earlier submission. The following is a list of the peer review reports and author responses from that submission.

Round 1

Reviewer 1 Report

In this paper Renz and colleagues perform a comprehensive search for novel transcriptional targets of FGF and TGFbeta signaling in hESCs. They do this by analysing effects of specific receptor inhibitors at shorter timings compared to previous reports. After elegant bioinformatic analysis, the search yielded a group of genes including STOX2, which was identified as a putative TGFbeta signaling pathway. The authors show that STOX2 expression is regulated by TGFbeta signaling and SMAD2 binds STOX2 regulatory regions. Finally, they show that STOX2 itself is necessary and sufficient for TGFbeta signaling responses and STOX2 bind SMADs.

Although the paper demonstrates throughout and rigorous analysis of the scientific question, which has clear developmental relevance, in order to be considered for publication in this journal, the manuscript requires significant changes.

Major concerns:

  • My main concern is the lack of mechanistic data to support the claim that STOX2 is a “cofactor” of SMAD2/4. Conversely, the manuscript only shows that STOX2 is a target of the pathway, and that STOX2 loss of function results in attenuated TGFbeta signaling. Although the authors show that STOX2 can bind SMADs, this is not proof that they have a functional interplay. For this the authors would need to:
    1. Determine whether the effect of STOX2 overexpression in Fig 5C on LEFTY2 and NODAL depend in SMAD activity? This dependency could also be tested using luciferase reporters of TGFbeta activity.
    2. Study whether overexpression of STOX2 stabilise SMAD expression
    3. Determine whether STOX2 binds LEFTY2 or NODAL promoters together with SMADs
  • As immunoprecipitation experiments in Figure 6 use whole cell lysates, the result cannot be interpreted as direct interaction. For that recombinant protein-based assays would be necessary

  • It is unclear to me why GFP-tagged STOX2 is not shown in the blots in figure 5D and these only show STOX2 around 100 KDa which is the size for endogenous (or untagged) STOX2. Could the authors please clarify whether cells were stably transfected with STOX2-GFP or with a bicystronic vector expressing STOX2 and GFP separately?

  • I would suggest re-organisation of the manuscript moving the developmental data (Figure 4) to the end, as at the moment that part is disconnected from the story. Regarding to this part, do the authors think that the re-expression of STOX2 during ectoderm differentiation is due to an increase in the binding of SMADs to the STOX2 promoter?

  • I would suggest removing Figure 6C as the quality of the result is not good enough to indicate STOX2-SMAD4 interaction. On the other hand, evidence for STOX2-SMAD2 interaction is solid.

Minor points:

  • Figure 2E is not mentioned in the manuscript and does not have a legend.
  • The way the data using the inhibitors in figure 1 is not consistent. Figure 1A shows SB-SU, then 1B: SU-SB, then 1E SB-SU. Please correct to avoid confusion for the readers
  • Figure legends should have a title.

Author Response

All changes have been highlighted in our revised manuscript.

Reviewer 1:

In this paper, Renz and colleagues perform a comprehensive search for novel transcriptional targets of FGF and TGFbeta signaling in hESCs. They do this by analyzing the effects of specific receptor inhibitors at shorter timings compared to previous reports. After elegant bioinformatic analysis, the search yielded a group of genes including STOX2, which was identified as a putative TGFbeta signaling pathway. The authors show that STOX2 expression is regulated by TGFbeta signaling and SMAD2 binds STOX2 regulatory regions. Finally, they show that STOX2 itself is necessary and sufficient for TGFbeta signaling responses and STOX2 binds SMADs.

We thank reviewer 1 for this constructive comment and have improved our manuscript accordingly in the 10 days given for the revision.

Although the paper demonstrates throughout and rigorous analysis of the scientific question, which has clear developmental relevance, in order to be considered for publication in this journal, the manuscript requires significant changes.

Major concerns:

  • My main concern is the lack of mechanistic data to support the claim that STOX2 is a “cofactor” of SMAD2/4. Conversely, the manuscript only shows that STOX2 is a target of the pathway and that STOX2 loss of function results in attenuated TGFbeta signaling. Although the authors show that STOX2 can bind SMADs, this is not proof that they have a functional interplay. For this the authors would need to:
    1. Determine whether the effect of STOX2 overexpression in Fig 5C on LEFTY2 and NODAL depends on SMAD activity? This dependency could also be tested using luciferase reporters of TGFbeta activity.
    2. Study whether overexpression of STOX2 stabilize SMAD expression
    3. Determine whether STOX2 binds LEFTY2 or NODAL promoters together with SMADs

We agree with the reviewer and tried to clarify this aspect in the revised manuscript. According to the short time given for the revision of our manuscript, we were not able to perform the proposed experiments. Nevertheless, we have amended our manuscript in this direction.

Finally, we would like to stress the fact that currently available antibody for STOX2 are not suitable to perform ChIP approaches.

  • As immunoprecipitation experiments in Figure 6 use whole-cell lysates, the result cannot be interpreted as direct interaction. For that recombinant protein-based assays would be necessary

We agree with the reviewer and clarify this aspect in the revised manuscript.

  • It is unclear to me why GFP-tagged STOX2 is not shown in the blots in figure 5D and these only show STOX2 around 100 KDa which is the size for endogenous (or untagged) STOX2. Could the authors please clarify whether cells were stably transfected with STOX2-GFP or with a bicistronic vector expressing STOX2 and GFP separately?

We thank the reviewer for raising this point and we now have improved our M&M section to clarify this point. The transfections were transient and STOX2 was GFP tagged. The acrylamide gel used for the blot shown in the former figure 5D was 12% acrylamide. In retrospective, it would have been more appropriate to use a low percentage gel (e.g.7.5%) or a gradient gel to better separate the 100kD untagged and 126kD tagged protein. Close observation of the blot in figure 5D shows that STOX2-GFP might migrates slightly higher than the untagged version.

  • I would suggest re-organization of the manuscript moving the developmental data (Figure 4) to the end, as at the moment that part is disconnected from the story. Regarding this part, do the authors think that the re-expression of STOX2 during ectoderm differentiation is due to an increase in the binding of SMADs to the STOX2 promoter?

We agree with the reviewer and have now moved the developmental data at the end of our manuscript (revised figure 6).

Concerning the question of whether the re-expression of STOX2 at later stages is due to increased binding of SMADs to the STOX2 enhancer: this is highly unlikely as the medium used for neurectoderm specification contains inhibitors to both TGF-b and BMP signaling. Indeed, phosphorylation of SMAD2 in the differentiation time series in the former figure 4C is only detectable in undifferentiated hESCs. Most likely the increase in STOX2 expression is caused by other signaling pathways such as WNT (see CTNNB1 blot in Figure 2b (beta-catenin)) binding to the STOX2 enhancer in figure 3 or SHH as proposed in the discussion.

  • I would suggest removing Figure 6C as the quality of the result is not good enough to indicate STOX2-SMAD4 interaction. On the other hand, evidence for STOX2-SMAD2 interaction is solid.

We agree with the reviewer and have removed this panel in our revised Figure 5.

Minor points:

  • Figure 2E is not mentioned in the manuscript and does not have a legend.

We do not have figure 2E in our manuscript, we nevertheless verified that all our figures are properly cited in our revised manuscript.

  • The way the data using the inhibitors in figure 1 is not consistent. Figure 1A shows SB-SU, then 1B: SU-SB, then 1E SB-SU. Please correct to avoid confusion for the readers

We thank the reviewer for noticing this inconsistency and have modified figure 1B in our revised manuscript.

  • Figure legends should have a title.

We now added titles to all the figure legends in our revised manuscript.

Reviewer 2 Report

FGF and TGFbeta pathways are key regulators of ESC maintenance and differentiation and previous work has focused on studying these pathways upon longer term treatment (ie 12-48 h).  Here, the authors seek to investigate early transcriptional changes (ie 3 – 9 h).  In the first part of the manuscript, the focus is pathway crosstalk.  Through a series of experimental and bioinformatic analysis, the results show that there are very few common direct targets.  In the second part of the manuscript, the authors select one TGFbeta target gene, STOX2 and show that STOX2 is not only a target gene but also a Smad2/4 partner.

Overall the authors should be commended on the quality of the experiments which were well controlled.  The the conclusions based on the experiments are sound and the work provides new knowledge in regards to the maintenance of pluripotency and differentiation of stem cells. 

A few minor comments are provided below that are not essential but would enhance the manuscript.

Minor comments:

Validation by qPCR of some of the direct common (TGFbeta/FGF) target genes would have been nice to see.

The author’s state that they did not observe STOX1 to be regulated by TGFbeta in their analysis.  Validation by qPCR would have provided greater confidence in this conclusion.

There are a few typos and English grammar errors.

Line 262:  ‘is still in its beginnings’ should be better worded

Line 306:  Sentence should not start with a number “16 of the 23....”

Line 405: migrated ‘trough’ should be ‘through’

Line 450:  ‘whereby’ should be ‘whereas’

Line 463:  ‘unspecific’ should be ‘nonspecific’

Author Response

Please see word document attach